



# FRESCO-B: A fast cloud retrieval algorithm using oxygen B-band measurements from GOME-2

Marine Desmons, Ping Wang, Piet Stammes, and L. Gijsbert Tilstra

Royal Netherlands Meteorological Institute (KNMI), De Bilt, The Netherlands

**Correspondence:** M.Desmons (desmons@knmi.nl)

**Abstract.** The FRESCO (Fast Retrieval Scheme for Clouds from the Oxygen A-band) algorithm is a simple, fast and robust algorithm used to retrieve cloud information in operational satellite data processing. It has been applied to GOME-1, SCIAMACHY, GOME-2 and more recently to TROPOMI. FRESCO retrieves effective cloud fraction and cloud pressure from measurements in the oxygen A-band around 761 nm. In this paper, we propose a new version of the algorithm, called FRESCO-B, which is based on measurements in the oxygen B-band around 687 nm. Such a method is interesting for vegetated surfaces where the surface albedo is much lower in the B-band than in the A-band, which limits the ground contribution to the top-of-atmosphere reflectances. In this study we first perform retrieval simulations. These show that the retrieved cloud pressures from FRESCO-B and FRESCO differ only between -10 hPa and +10 hPa, except for high thin clouds over vegetation where the difference is larger, about +15 to +30 hPa, with FRESCO-B yielding higher pressures. Next, inter-comparison between FRESCO-B and FRESCO retrievals over one month of GOME-2B data reveals that the effective cloud fractions retrieved in the $O_2$ A and B bands are very similar (mean difference of 0.003) while the cloud pressures show a mean difference of 11.5 hPa, with FRESCO-B retrieving higher pressures than FRESCO. This agrees with the simulations and is partly due to deeper photons penetrations of $O_2$ B-band in clouds as compared to the $O_2$ A-band photons, and partly due to the surface albedo bias in FRESCO. Finally, validation with ground-based measurements shows that the FRESCO-B cloud pressure represents an altitude within the cloud boundaries for clouds that are not too far from the Lambertian reflector model, which occurs in about 50% of the cases.

*Copyright statement.* TEXT

## 1 Introduction

The Global Ozone Monitoring Experiment-2 (GOME-2) is a spectrometer flying on the Metop series of satellites: on Metop-A since 2006, on Metop-B since 2012, and on Metop-C, launched in November 2018. The GOME-2 instruments sense the Earth back-scattered radiance and solar irradiance in the ultraviolet and visible part of the spectrum (240-790 nm) with a spectral resolution between 0.26 nm and 0.51 nm. The primary goal of GOME-2 measurements is the study of ozone as well as atmospheric trace gases and pollutants (e.g. nitrogen dioxide, sulfur dioxide, water vapor, bromine oxide). The trace



gas retrieval algorithms rely on information on cloud properties for each ground pixel. Indeed, clouds strongly affect trace gas retrievals from satellite measurements because of their shielding effect, their albedo effect and because of the in-cloud absorption effect (Stammes et al., 2008). Therefore, in order to ensure good quality trace gas retrievals, it is very important to retrieve cloud properties, and in particular, the cloud top height, cloud geometrical thickness, cloud fraction, cloud optical

thickness and the number of cloud layers (Boersma et al., 2004).

From a larger perspective, clouds are a key component of the Earth's climate system through their role in the Earth hydrological cycle and radiation balance. Global observation and description of clouds are necessary to understand and properly depict their overall and multiple effects. This is particularly true in the context of the climate change we are experiencing. Especially, the question whether the coverage of different cloud types will change, or the partition of low-level versus high-level clouds -

which have different and sometimes opposite radiative effects - will change is a crucial one. This is recognized as one of the major challenges in climate change predictions (Bony and Dufresne, 2005; Andrews et al., 2012; Vial et al., 2013).

The usage of the oxygen absorption for the retrieval of cloud pressure has already been studied for several decades. Indeed, $O_2$ is well mixed in the atmosphere and the degree of $O_2$ absorption can be related to a certain atmospheric path length. Above a bright surface, as a cloud acts in first approximation, $O_2$ absorption that affects solar radiation back-scattered towards a

space-borne sensor is mainly related to the scene height (the cloud height in our case). Such methods using reflected sunlight in oxygen absorption bands depend very weakly on the pressure/temperature vertical profiles. They do not suffer from a lack of sensitivity in the case of low clouds, and are not sensitive to temperature inversions, like retrievals with infrared measurements. The use of the oxygen A-band for remote sensing of cloud properties has been extensively studied (Wu, 1985; Fischer and Grassl, 1991; Kuze and Chance, 1994), and used in airborne campaigns (Fischer et al., 1991) and satellite missions

(Vanbauce et al., 1998; Koelemeijer et al., 2001; Fournier et al., 2006; Lindstrot et al., 2006; Preusker et al., 2007; Lelli et al., 2012), demonstrating the capabilities to retrieve an apparent cloud pressure using different sensors with narrow spectral bands centered on the oxygen A-band region. However, the use of the oxygen B-band for such retrievals remains quite limited, usually in association with measurements in the A-band (Kuze and Chance, 1994; Yang et al., 2013), or applied to the retrieval of aerosols or vegetation properties (Marshak and Knyazikhin, 2017; Xu et al., 2017). In this paper, we propose a new version

of the FRESCO algorithm, called FRESCO-B, which is based on measurements in the B-band. Such a method is interesting for vegetated surfaces. Indeed for this type of surface, the surface reflectance is much lower in the B-band than in the A-band (Tilstra et al., 2017), limiting the ground contribution to the top-of-atmosphere (TOA) measured reflectances.

This paper is organised as follows. In Sect. 2, we present the FRESCO algorithm and our motivations to apply it in the oxygen B-band. In Sect. 3, we describe the oxygen A and B-bands, as well as the FRESCO-B retrieval method. In Sect. 4, we

perform simulations of cloud retrievals in the oxygen A and B bands. In Sect. 5, we apply FRESCO-B to GOME-2 data, and validate the results by performing comparisons with FRESCO and with ground-based measurements. We conclude this study in Sect. 6.



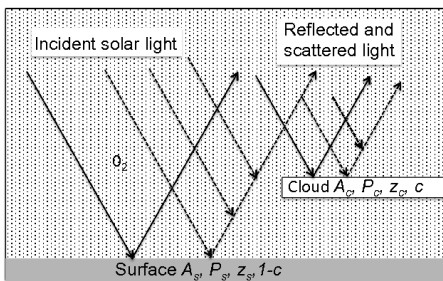

**Figure 1.** Atmospheric radiation model used in the FRESCO retrieval algorithm. Three light paths are considered: (1) from sun to surface to satellite, (2) from sun to cloud to satellite (1-2: solid lines), (3) from sun to atmosphere to satellite according to single Rayleigh scattering (dashed lines). The pixel is partly cloudy, with $c$ the effective cloud fraction, and partly clear. $A_s$, $P_s$ and $z_s$ stand respectively for the surface albedo, pressure and altitude, while $A_c$, $P_c$ and $z_c$ indicate respectively the cloud albedo, pressure and altitude.

## 2 The FRESCO algorithm

In the FRESCO algorithm (Koelemeijer et al., 2001; Wang et al., 2008), information on cloud pressure and effective cloud fraction is derived from the reflectances in three 1 nm wide windows, situated in and around the $O_2$ A-band. The algorithm fits a simulated reflectance spectrum to the measured reflectance spectrum in the three windows, namely 758-759 nm, 760-761 nm and 765-766 nm. The atmospheric model used, shown in Figure 1, is based on the independent pixel approximation: the top-of-atmosphere simulated reflectances ($R_{sim}$) are computed as the weighted sum of the reflectances of the cloud-free and cloudy parts of the pixel, using the cloud fraction for the weight. The atmosphere above the ground surface or cloud is treated as an absorbing (due to oxygen) and purely Rayleigh scattering medium. The simulated reflectances can be written as (Wang et al., 2008):

$$R_{sim} = (1-c)T_s A_s + (1-c)R_s + cT_c A_c + cR_c \qquad (1)$$

where $c$ is the effective cloud fraction while $A_c$ and $A_s$ stand for the cloud and surface albedo. $R_c$, $T_c$ and $R_s$, $T_s$ are the single Rayleigh scattering reflectances and transmittances of the cloudy and cloud-free part of the pixel, respectively. The transmission and Rayleigh scattering reflectances are pre-calculated as a function of the solar zenith angle (SZA), viewing zenith angle (VZA), azimuth difference, wavelength and pressure level ($P_c$, $P_s$). The transmission and reflectance spectra are calculated using a line-by-line method using the line parameters from HITRAN 2016 (Gordon et al., 2017) and then convolved using the instrument spectral response function at the measurement wavelength grid. In this model, reflection occurs only at the ground surface or cloud top, and both the ground surface and the cloud are assumed to be Lambertian reflectors. Consequently, the area below the cloud does not contribute in the radiative transfer calculation. The surface albedo is taken from a climatology (Tilstra et al., 2017) while the surface pressure is calculated from surface elevation, using the Mid-Latitude Summer atmospheric profile. The cloud albedo is assumed to be 0.8 or set to the reflectance at 758 nm if the reflectance is larger





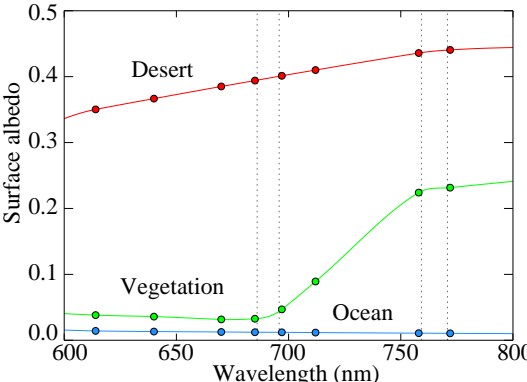

**Figure 2.** Mean surface albedo over one month (August) and for different surface types as a function of wavelength. "Desert" stands for the Sahara desert, "Ocean" for the Atlantic ocean, and "Vegetation" for the Amazonian tropical rain forest. The vertical dashed lines indicate the location of the oxygen A band (around 761 nm) and B band (around 687 nm). The values come from the SCIAMACHY surface albedo database described in Tilstra et al. (2017).

than 0.8. The retrieval method is based on minimizing the difference between the measured and simulated spectrum, using a Levenberg-Marquardt nonlinear least-squares method.

The primary aim of FRESCO is to serve the cloud correction in the trace gas retrievals, but cloud modellers are also in-terested in the FRESCO data, because the retrieval method uses oxygen and not temperature, and is therefore also sensitive
5   to low, warm clouds. The FRESCO cloud algorithm is simple, fast and robust (Wang and Stammes, 2014) and therefore suit-able for operational processing; it has been applied to GOME-1, SCIAMACHY, GOME-2 and more recently, to TROPOMI. Consequently, it is worth continuing to improve the method and to develop new applications.

A motivation to use the B-band to estimate the cloud height is the surface albedo at the B-band wavelengths. Figure 2 shows the mean surface albedo values from the SCIAMACHY surface Lambertian-Equivalent Reflectivity (LER) database
10   (Tilstra et al., 2017). The values are taken for the month of August, for different regions (Atlantic ocean, Sahara desert and Amazonian Tropical rain forests), and is of the "MODE-LER" type. We can see that for vegetation, the surface albedo is significantly lower within the B-band than within the A-band. This means that the contribution of the ground in the top-of-atmosphere reflectances is lower in the oxygen B-band than in the A-band which may lead to more accurate cloud properties retrieval in the B-band over vegetation.





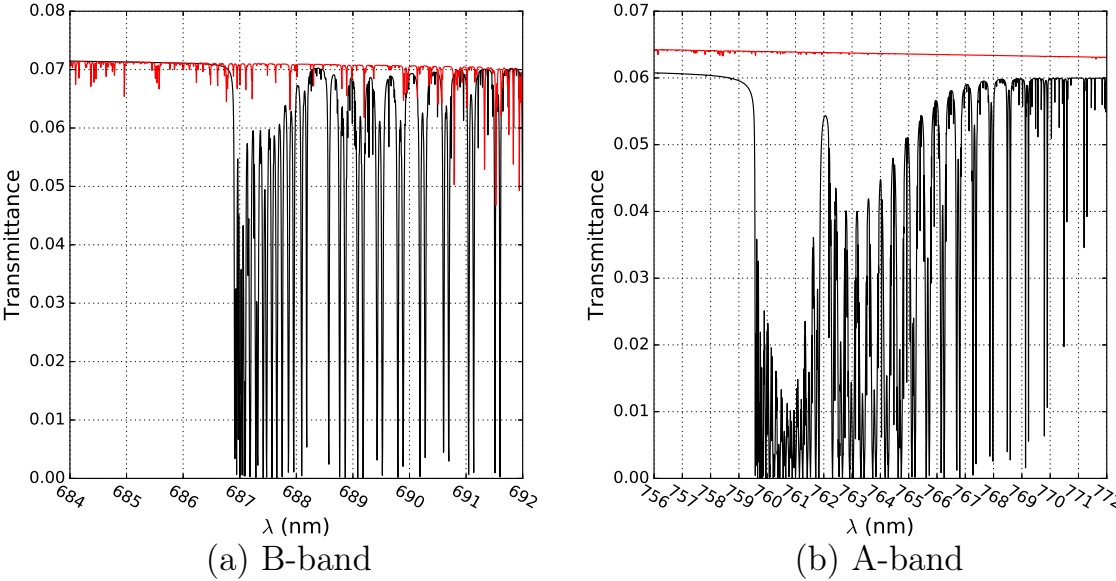

**Figure 3.** Line-by-line transmittances in the oxygen A (b) and B (a) bands (black lines). On panel (a), the transmittance of the overlapping water vapor lines is represented in red. The transmittances are computed using the HITRAN 2016 database (Gordon et al., 2017).

## 3 Radiative transfer in oxygen A and B-bands and FRESCO-B retrieval method

### 3.1 Oxygen A and B bands

The oxygen molecule has two strong absorption bands: the A-band around 761 nm and the B-band around 687 nm. Figure 3 shows the high-resolution transmittances, calculated using the 2016 HITRAN database (Gordon et al., 2017). We can clearly

5   see that the B-band is less deep than the A-band, which means that less light is absorbed by oxygen in the B-band than in the A-band. With the hypothesis of a cloud acting like a high-albedo Lambertian reflector, the cloud height retrieved in the B-band is usually lower than the cloud height retrieved in the A-band (Yang et al., 2013). This is visible in Figure 4 which shows GOME-2B measurements in the oxygen A and B bands for two clouds at different altitudes over ocean. For the low cloud (black lines), FRESCO-B and FRESCO retrieves similar cloud pressures of 975 hPa and 980 hPa respectively. However, for the high cloud,

10  the retrieved pressures are quite different as FRESCO-B cloud pressure is 461 hPa while FRESCO cloud pressure is 412 hPa. This large difference will be discussed later on but it is clear that retrieving cloud height from measurements in the oxygen B-band is very valuable. Indeed, it brings new information regarding the vertical location of the cloud, which can then be used together with the measurements in the oxygen A-band in order to have more information about the cloud layer.





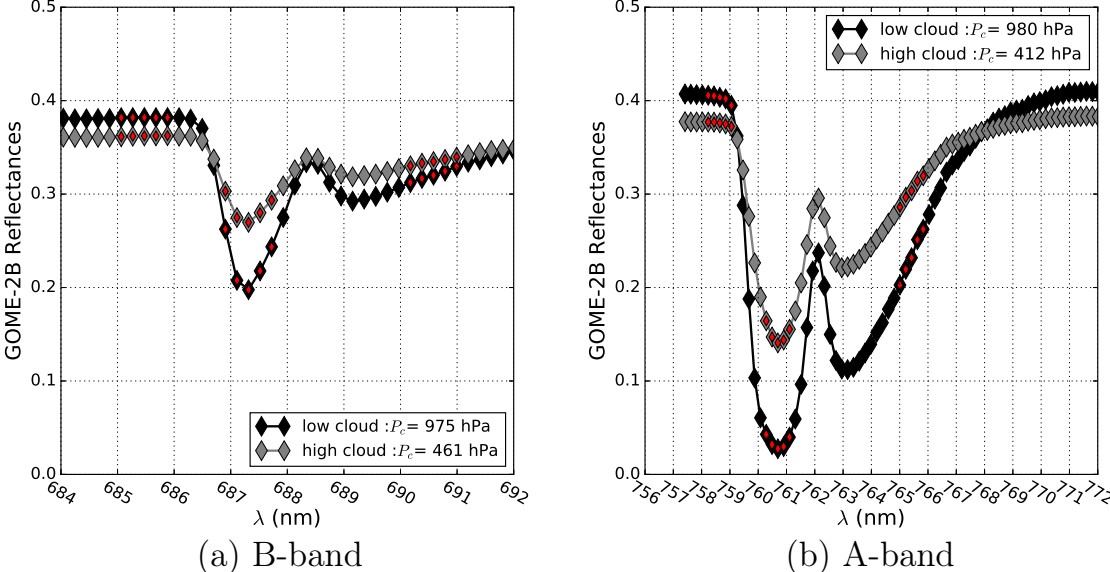

**Figure 4.** Example of GOME-2B spectral reflectance measurements in the B-band (a) and A-band (b) for low clouds (black line) and high clouds (grey line) for a pixel over ocean. The red colored symbols indicate the measurements that are used in the FRESCO-B and FRESCO algorithms, respectively. The cloud pressures retrieved with FRESCO-B and FRESCO are indicated.

## 3.2 FRESCO-B retrieval method

Similarly to FRESCO (Koelemeijer et al., 2001; Wang et al., 2008), the FRESCO-B retrieval method is based on minimizing the difference between a measured and a simulated spectrum using a Levenberg-Marquardt nonlinear least-squares method. FRESCO-B retrieves the effective cloud fraction as well as the cloud pressure from the top-of-atmosphere reflectances at three 1 nm wide windows, namely 685-686 nm, 686.8-687.8 nm, and 690-691 nm. The wavelengths are chosen in order to maximize the difference of absorption between the windows. Also, Figure 3 shows that the contamination by water vapor is small so we neglect it. Each of the three windows contains 5 reflectance measurements by GOME-2B as can be seen on Figure 4a. While the reflectances measured in the continuum (between 685 nm and 686 nm) are not impacted by the altitude of the cloud but only by its albedo, the amount of absorption in the two other windows varies according to the cloud altitude.

## 4 Simulations

### 4.1 Methodology

In order to understand FRESCO-B and its differences with FRESCO, we have simulated spectra for four different cloud cases. $O_2$ A and B-band reflectance spectra have been simulated with the DAK (Doubling-Adding KNMI) model (De Haan et al.,





1987; Stammes et al., 1989; Stammes, 2001), which is a line-by-line radiative transfer model. The spectral simulations have been made for a Mid-Latitude Summer atmosphere consisting of 33 plane-parallel homogeneous layers with multiple scattering and oxygen absorption. The $O_2$ absorption cross-sections were calculated using HITRAN 2016 line parameters (Gordon et al., 2017). In this atmosphere, homogeneous scattering cloud layers are inserted, with varying optical thickness. The cloud particle

scattering phase function is a Henyey-Greenstein function with an asymmetry parameter of 0.85 and a single scattering albedo of 1. The cloud scenes are simulated with single-layer clouds which fully cover the pixels with a top altitude of either 5 km or 10 km, a geometrical thickness of 3 km, and for an optical thickness of 9 or 42. The four different cases, similar to the ones used in Sneep et al. (2008), are described in Table 1. The spectra were calculated from 684 nm to 692 nm for the B-band and from 756 nm to 772 nm for the A-band, both at 0.005 nm resolution. For each band, the simulations have been done for

ocean and vegetation, with surface albedo values taken from the database described in Sect. 2. Then, we have convolved the obtained spectra with the GOME-2B slit functions and applied the FRESCO-B and FRESCO algorithms. Results are shown for 4 different viewing and solar geometries.

## 4.2   Cloud fraction

In Figure 5a, we can see that for the cloud cases over ocean, the effective cloud fraction retrieved with FRESCO-B and

FRESCO are very similar, as the difference ranges from 0 to 0.02. Figure 4b shows that over vegetation, the difference between the two effective cloud fractions is higher as it is comprised between 0.01 and 0.04. The effective cloud fraction retrieved with FRESCO-B is always higher than the one retrieved with FRESCO. This behaviour is as expected. Indeed, for wavelengths in the continuum (which are used in the effective cloud fraction retrievals), we may set $T = 1$ in Equation 1. Using $R_{sim}(\lambda) \approx R_{meas}(\lambda)$ leads to:

$$c = \frac{R_{meas} - A_s - R_s}{A_c - A_s + R_c - R_s} \tag{2}$$

Differentiation to $A_s$ gives the change in the retrieved effective cloud fraction, $\Delta c$, due to a small change in the assumed surface albedo, $\Delta A_s$ (Koelemeijer et al., 2001):

$$\Delta c = -\frac{1-c}{A_c - A_s + R_c - R_s} \Delta A_s \tag{3}$$

As the albedo of vegetation is lower in the B-band than in the A-band (see Fig. 2), we expect to retrieve a higher cloud fraction

with FRESCO-B for this type of surface. For ocean, the surface albedo chosen for the simulations is the same in the two spectral regions, so we do not expect any impact on the effective cloud fraction.

## 4.3   Cloud pressure

In Figure 6, we see that the pressures retrieved by FRESCO-B and FRESCO in the simulations, indicate an altitude inside the cloud layer but well below the cloud top altitude. This feature is well known and common to algorithms which are based upon

the hypothesis of a Lambertian cloud to retrieve the cloud top pressure (Saiedy et al., 1965; Vanbauce et al., 1998; Parol et al., 1999; Wang et al., 2008; Sneep et al., 2008). Indeed, real clouds do not act as perfect reflecting boundaries and the algorithm



| | Simulated cloud cases | | | |
|---|---|---|---|---|
| | 1 | 2 | 3 | 4 |
| Description | thin and low | thick and low | thin and high | thick and high |
| Cloud top | 5 km | 5 km | 10 km | 10 km |
| Cloud bottom | 2 km | 2 km | 7 km | 7 km |
| Total cloud optical thickness | 9 | 42 | 9 | 42 |
| Surface albedo | Ocean | Vegetation | | |
| Oxygen A-band | 0.02 | 0.2 | | |
| Oxygen B-band | 0.02 | 0.05 | | |

**Table 1.** Parameters for the four cloud cases used in the retrieval simulations.

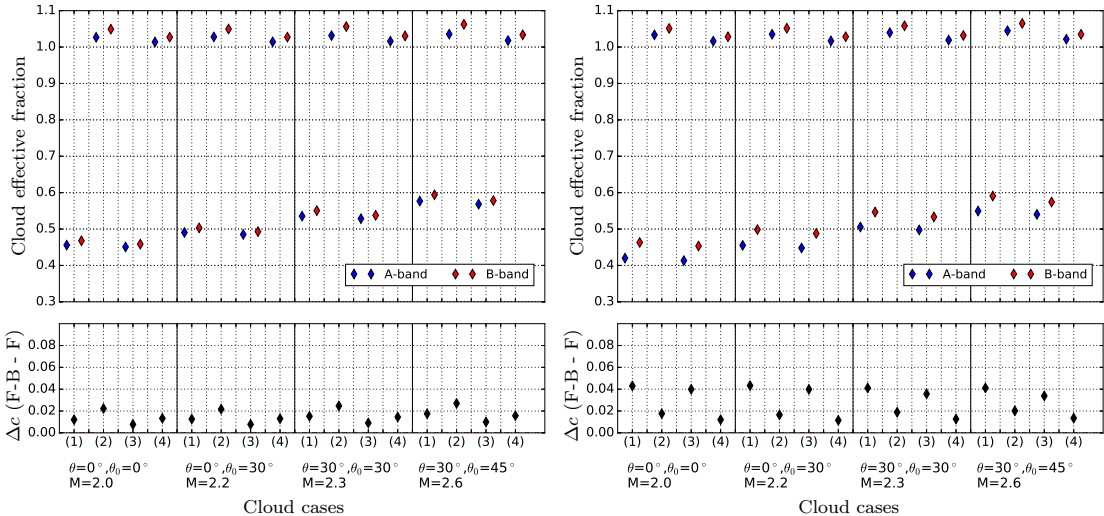

**Figure 5.** FRESCO-B (F-B) and FRESCO (F) effective cloud fraction retrievals (top panels) and effective cloud fraction differences (bottom panels) for the simulated cloud scenes over ocean (a) and vegetation (b). The cloud cases are described in Table 1. The simulations have been made with an azimuth angle difference of $90°$.

takes into account neither the photons reflected by the surface below the cloud, nor the photons penetration inside the cloud layer. In both cases, the photons path increases, as well as the oxygen absorption, which leads to a higher pressure than the cloud top. For optically thick clouds, the retrieved pressure is closer to the cloud top than for thinner clouds; the thick clouds





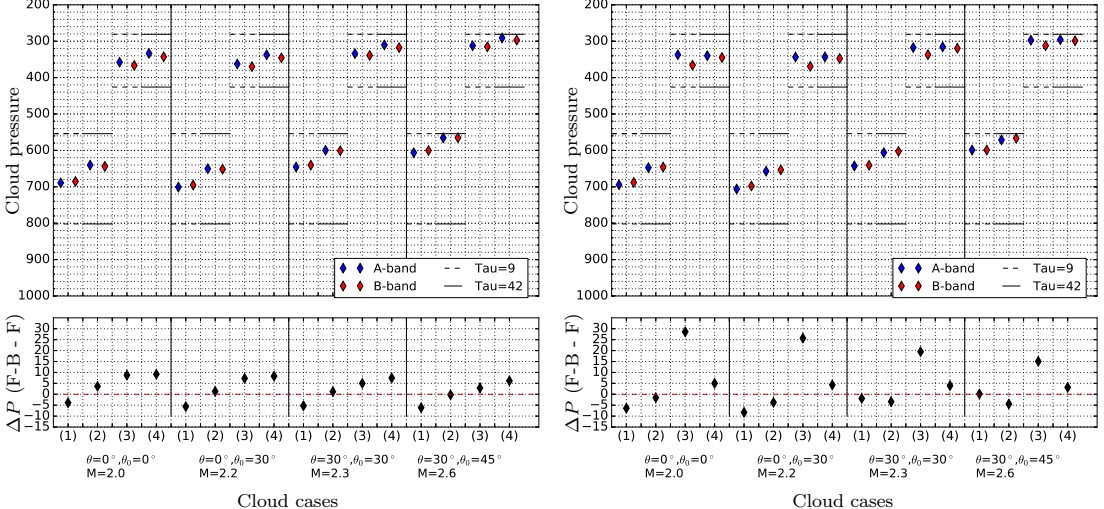

**Figure 6.** FRESCO-B (F-B) and FRESCO (F) cloud pressure retrievals (top panels) and cloud pressure differences (bottom panels) for the simulated cloud scenes over ocean (a) and vegetation (b). The cloud cases are described in Table 1. The simulations have been made with an azimuth angle of $90°$.

are closer to the model of a Lambertian cloud. We also observe a decrease in the retrieved cloud pressures with increasing geometric air mass factor. Indeed, the larger the solar and viewing zenith angles are, the shallower the photon penetrates inside the cloud layer.

In Figure 6a, we can see that, over ocean, the difference in retrieved pressure between FRESCO-B and FRESCO is between about $-5\,\mathrm{hPa}$ and $+10\,\mathrm{hPa}$, and increases with the cloud altitude. Indeed, the higher the cloud, the longer the photon's path under the cloud, where it undergoes oxygen absorption. As this path under the cloud is not taken into account in FRESCO and FRESCO-B, high cloud altitudes lead to larger pressure differences.

In Figure 6b, we can see that over vegetation, the difference of pressures between FRESCO-B and FRESCO is between about $-10\,\mathrm{hPa}$ and $+10\,\mathrm{hPa}$, excepted for high (top altitude $10\,\mathrm{km}$) optically thin clouds ($\tau = 9$) clouds where the difference is much larger. The reason that FRESCO-B retrieves a (much) higher cloud pressure than FRESCO for high (thin) clouds is twofold. Firstly, the $O_2$ B-band is less strong than the $O_2$ A-band, so that radiation in the B-band penetrates deeper into clouds and the atmosphere than in the A-band, thus down to higher pressures. Secondly, for optically thin clouds there is a relatively large cloud-free part of the pixel in the FRESCO retrieval model (see Fig. 1). Since the surface albedo of land is higher in the $O_2$ A-band than in the $O_2$ B-band, there are more photons reflected by the surface in the A-band than in the B-band for the cloud-free part. These photons have experienced a high pressure and a large $O_2$ absorption. To compensate for this large $O_2$ absorption, the FRESCO retrieval places the cloud at a lower pressure than the FRESCO-B retrieval does. Both effects lead to a positive difference between FRESCO-B and FRESCO cloud pressures. The second effect explains the surface albedo bias of the FRESCO retrieval as compared to the FRESCO-B retrieval.





The FRESCO algorithm works best for clouds over a dark surface, because then the major part of the radiation is coming from the cloud and not from the surface. When the surface albedo is increased in FRESCO, which happens when going from the $O_2$ B-band to the $O_2$ A-band, the cloud fraction decreases and the cloud pressure decreases as well (the cloud rises). This behaviour agrees with earlier studies, simulations and retrievals of FRESCO using the A-band when changing the surface

albedo. In our simulations of clouds over vegetation the cloud fraction is indeed decreasing when going from the B-band to the A-band, and the cloud pressure is decreasing for high thin clouds by about 25-30 hPa when going from the B-band to the A-band.

## 5 Results

### 5.1 FRESCO in the oxygen A and B bands applied to GOME-2B data

In this section we compare the effective cloud fraction and cloud pressure retrieved by the two versions of FRESCO. We have run the two algorithms over one month of GOME-2B data in July 2014, excluding the snow and ice cases. The dataset contains 4 208 125 cases.

### 5.1.1 Cloud fraction

The differences of effective cloud fraction between FRESCO-B and FRESCO are shown of Figure 7a. The distribution is

centered around 0 and has a mean value of 0.003 and a standard deviation of 0.036. We see the highest difference at high latitudes and over bright surfaces. Table 2 shows the mean effective cloud fraction retrieved in the oxygen A and B bands for different kinds of surface. We can see that the differences are almost zero, which is expected as the cloud fraction is mainly determined through the reflectance measurements performed in the non-absorbed part of the spectrum.

However, over vegetation, both Figure 7a and Table 2 show that the effective cloud fraction retrieved in the B-band is lower

than the one retrieved in the A-band while the simulations suggest the opppsite (cf. section 4.2). Indeed, although the surface reflectance has an anisotropic BRDF function, the surface is often assumed to be Lambertian, as in many situations the full BRDF is not available or the radiative transfer code used is not equipped to handle the BRDF properly. This is the case in FRESCO and FRECO-B for which we use a MODE-LER surface albedo climatology established from SCHIAMACHY measurements (Tilstra et al., 2017). This anisotropy is stronger over vegetation which has non-isotropic elements like dense

tree with heterogeneous leaves and shadow effects) and in the NIR (0.7-2.5 µm) where the atmosphere is more transparent. Recently, (Lorente et al., 2018) has shown that this assumption of a Lambertian surface, leads to across-track biases on satellite retrievals that use those climatologies, like the effective cloud fraction, for the solar and viewing geometries of GOME-2. It is shown by Lorente et al. (2018), that the western part of the swath has the most biased FRESCO cloud fraction. Consequently, for the vegetation case, we have recompiled the mean effective cloud fractions for FRESCO and FRESCO-B, distinguishing

the eastern, nadir and western pixels of the GOME-2B swath. The results are summarised in Table 3. We see that for the eastern and nadir pixels of the swath, the effective cloud fraction retrieved by FRESCO-B is higher than the one retrieved by





| | Nr. of | Effective cloud fraction | | Difference |
| | cases | FRESCO-B | FRESCO | FRESCO-B - FRESCO |
|---|---|---|---|---|
| All cases | 4 208 125 | $\bar{c} = 0.389$, SD=0.304 | $\bar{c} = 0.387$, SD=0.297 | $\overline{\Delta c} = +0.0027$, SD=0.036 |
| Ocean | 2 582 287 | $\bar{c} = 0.395$, SD=0.298 | $\bar{c} = 0.388$, SD=0.294 | $\overline{\Delta c} = +0.0066$, SD=0.023 |
| Land | 1 447 322 | $\bar{c} = 0.374$, SD=0.312 | $\bar{c} = 0.379$, SD=0.302 | $\overline{\Delta c} = -0.0052$, SD=0.051 |
| Vegetation | 920 902 | $\bar{c} = 0.380$, SD=0.320 | $\bar{c} = 0.390$, SD=0.310 | $\overline{\Delta c} = -0.0106$, SD=0.064 |

**Table 2.** Mean effective cloud fractions and standard deviations for FRESCO-B and FRESCO, as well as the mean differences, for GOME-2B measurements in July 2014 over different surfaces. The Vegetation category is a sub-part of the Land one.

| Vegetation | Nr. of | Effective cloud fraction | | Difference |
| | cases | FRESCO-B | FRESCO | FRESCO-B - FRESCO |
|---|---|---|---|---|
| All swath | 920 902 | $\bar{c} = 0.380$, SD=0.320 | $\bar{c} = 0.390$, SD=0.310 | $\overline{\Delta c} = -0.0106$, SD=0.064 |
| West | 269 861 | $\bar{c} = 0.396$, SD=0.322 | $\bar{c} = 0.440$, SD=0.290 | $\overline{\Delta c} = -0.0443$, SD=0.078 |
| Nadir | 286 768 | $\bar{c} = 0.342$, SD=0.311 | $\bar{c} = 0.338$, SD=0.304 | $\overline{\Delta c} = +0.0042$, SD=0.050 |
| East | 364 273 | $\bar{c} = 0.396$, SD=0.322 | $\bar{c} = 0.394$, SD=0.323 | $\overline{\Delta c} = +0.0026$, SD=0.053 |

**Table 3.** Mean effective cloud fractions and standard deviations for FRESCO-B and FRESCO, as well as the mean differences, for GOME-2B measurements in July 2014 over vegetation. We distinguish the eastern (pixels 1 to 8), nadir (pixels 9 to 16) and western (pixels 17 to 24) parts of the swath .

FRESCO, which agrees with our simulations, while for the western pixels, this is the opposite. Indeed, for the western part of the swath, the cloud effective fraction retrieved in the $O_2$ A-band is too high because the surface albedo used is high (red edge (Tilstra et al., 2017)) and the anisotropy is stronger, while in the $O_2$ B-band, the surface albedo is low and the anisotropy is smaller. Consequently, the error on $A_s$ due to the assumption of a Lambertian surface has a smaller impact on effective cloud

5 fraction in the B-band than in the A-band. This difference of behaviour of the retrievals according to the part of the swath corroborates the conclusions of Lorente et al. (2018).

### 5.1.2 Cloud pressure

In this subsection, we limit our study to the pixels for which the FRESCO effective cloud fraction is higher than or equal to 0.1, indeed, as mentioned in Wang et al. (2008), cloud pressures have big uncertainties when effective cloud fraction is lower

10 than 0.1. This selection leaves us with 3 237 790 pixels.





| | Nr. of cases | Cloud pressure (hPa) | | Difference (hPa) |
|---|---|---|---|---|
| | | FRESCO-B | FRESCO | FRESCO-B - FRESCO |
| All cases | 3 237 790 | $\overline{P} = 747$, SD=176 | $\overline{c} = 736$, SD=195 | $\overline{\Delta P} = 11.5$, SD=44.9 |
| Ocean | 2 032 709 | $\overline{P} = 763$, SD=179 | $\overline{c} = 749$, SD=199. | $\overline{\Delta P} = 13.9$, SD=42.1 |
| Land | 1 063 650 | $\overline{P} = 715$, SD=165 | $\overline{c} = 709$, SD=183 | $\overline{\Delta P} = 6.31$, SD=49.1 |
| Vegetation | 651 903 | $\overline{P} = 689$, SD=157 | $\overline{c} = 681$, SD=174 | $\overline{\Delta P} = 8.52$, SD=50.2 |

**Table 4.** Mean cloud pressures and standard deviations for FRESCO-B and FRESCO, as well as the mean differences, for FRESCO-B and FRESCO for GOME-2B measurements in July 2014 over different surfaces. The Vegetation category is a sub-part of the Land one.

| | Nr. of cases | Chi-squared | | Difference |
|---|---|---|---|---|
| | | FRESCO-B | FRESCO | FRESCO-B - FRESCO |
| All cases | 3 237 790 | $\overline{\chi_2} = 10.3$, SD=9.7 | $\overline{\chi_2} = 5.8$, SD=8.2 | $\overline{\Delta \chi_2} = 4.5$, SD=9.1 |
| Ocean | 2 032 709 | $\overline{\chi_2} = 10.1$, SD=10.0 | $\overline{\chi_2} = 5.5$, SD=8.1 | $\overline{\Delta \chi_2} = 4.7$, SD=9.2 |
| Land | 1 063 650 | $\overline{\chi_2} = 10.3$, SD=9.0 | $\overline{\chi_2} = 6.4$, SD=8.3 | $\overline{\Delta \chi_2} = 3.9$, SD=8.8 |
| Vegetation | 651 903 | $\overline{\chi_2} = 9.72$, SD=9.0 | $\overline{\chi_2} = 6.8$, SD=8.4 | $\overline{\Delta \chi_2} = 2.9$, SD=8.6 |

**Table 5.** Mean Chi-squared and standard deviations for FRESCO-B and FRESCO, as well as the mean differences for FRESCO-B and FRESCO for GOME-2B measurements in July 2014 over different surfaces. The Vegetation category is a sub-part of the Land one.

| Vegetation | Nr. of cases $c_{\mathrm{eff}} > 0.1$ | Cloud pressure (hPa) | | Difference (hPa) |
|---|---|---|---|---|
| | | FRESCO-B | FRESCO | FRESCO-B - FRESCO |
| All swath | 651 903 | $\overline{P} = 689$, SD=157 | $\overline{P} = 681$, SD=174 | $\overline{\Delta P} = 8.52$, SD=50.2 |
| West | 205 093 | $\overline{P} = 699$, SD=154 | $\overline{P} = 706$, SD=173 | $\overline{\Delta P} = -7.19$, SD=54.1. |
| Nadir | 185 336 | $\overline{P} = 705$, SD=153 | $\overline{P} = 689$, SD=171 | $\overline{\Delta P} = +16.2$, SD=46.6 |
| East | 261 474 | $\overline{P} = 671$, SD=161 | $\overline{P} = 655$, SD=174 | $\overline{\Delta P} = +15.4$, SD=46.6 |

**Table 6.** Mean cloud pressure for FRESCO-B and FRESCO, as well as differences, for GOME-2B measurements in July 2014 over vegetation. We distinguish the eastern (pixels 1 to 8), nadir (pixels 9 to 16) and western (pixels 17 to 24) parts of the swath .





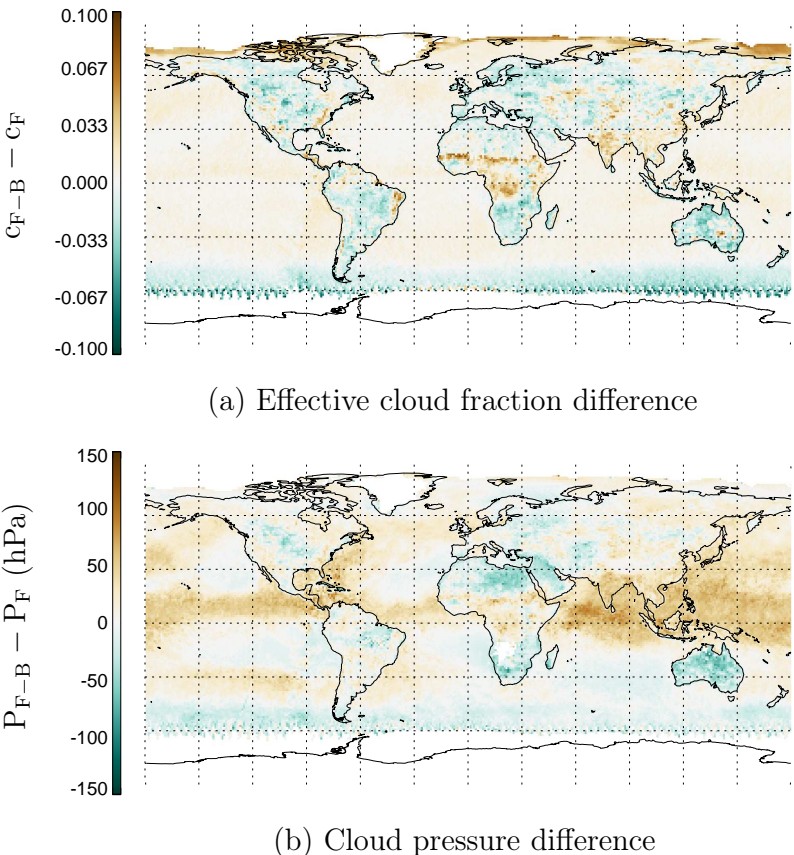

(a) Effective cloud fraction difference

(b) Cloud pressure difference

**Figure 7.** Differences of effective cloud fraction and cloud pressure between FRESCO-B and FRESCO for July 2014.

The differences of cloud pressures retrieved with FRESCO-B and FRESCO are shown on panel Figure 7b. The mean cloud pressures are $736 \pm 195 \, \mathrm{hPa}$ in the A-band and $747 \pm 176 \, \mathrm{hPa}$ in the B-band. As already mentioned, this behaviour is expected as the B-band is less absorbing. We also have analyzed the differences separating the pixels according to the underlying surface type. The mean cloud pressures as well the pressure differences between FRESCO-B and FRESCO for each surface are shown in Table 4.

Over ocean, the mean cloud pressure is $763 \pm 179 \, \mathrm{hPa}$ with FRESCO-B, while with FRESCO, this value is $749 \pm 199 \, \mathrm{hPa}$. The surface albedo of ocean is very similar in oxygen A and B bands, therefore the differences we observe in cloud pressures are only due to a difference of absorption inside and under the cloud. For instance, on Figure 7b, we observe the larger pressure differences in the ITCZ, where there are a lot of high clouds (cumulonimbus, cirrus), which agrees with the simulations presented in section 4.3.

Over vegetation, we observe on average small pressure differences, the mean cloud pressures being $689 \pm 157 \, \mathrm{hPa}$ with FRESCO-B and $681 \pm 174 \, \mathrm{hPa}$ with FRESCO. Figure 8 allows us to visualize the vegetated areas in July (the higher the



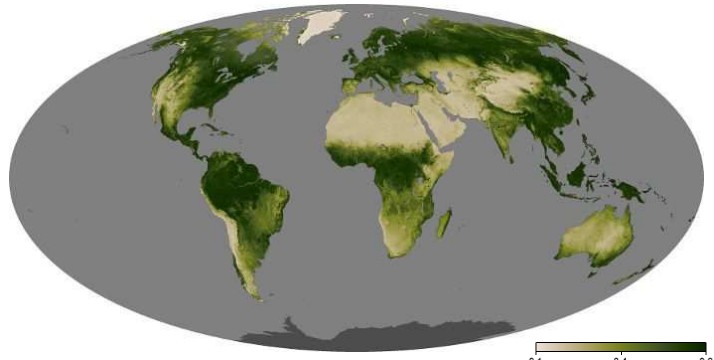

**Figure 8.** MODIS NDVI index for July 2014. Source: $https://earthobservatory.nasa.gov/global-maps/TRMM_3B43M/MOD_NDVI_M$

index, the greener the area). We can see on Figure 7b that those areas (North-West of North America, North of South America, Northern part of Europe and Asia) have, on average, small cloud pressures differences, which also agrees with the simulations we have described in Section 4.3. Moreover, as seen on Figure 2, the surface albedo of vegetation surfaces is much lower in the oxygen B-band than in the oxygen A-band. Therefore, the contribution of the ground in the reflectances is lower in the

oxygen B-band than in the A-band, so we expect more accurate retrievals in the B-band. Table 5 shows the value of $\chi^2$ obtained with FRESCO and FRESCO-B for different types of surfaces. We can see that the $\chi^2$ are always higher for FRESCO-B than FRESCO, this is due to the fits of the algorithms: the difference between the simulated reflectances and the measured ones is always higher in the B-band. However, we can notice that while with FRESCO the $\chi^2$ is the higher for vegetation, this is the opposite with FRESCO-B. FRESCO-B is the most suited over vegetated surfaces.

As we have previously mentioned the anisotropy of the surface albedo over vegetation and its consequences on effective cloud fraction retrieval, we have compiled the mean cloud pressure retrieved with FRESCO and FRESCO-B for the different part of the swath. The results are summarized in table 6. We see that the pressure retrieved with FRESCO-B is higher than the one retrieved with FRESCO for the eastern and nadir part of the swath, while it is the opposite for the western pixels. This last observation is due to the bias albedo on the western part of the swath: As mentioned in section 5.1.1, the use of a

Lambertian albedo in FRESCO and FRESCO-B leads to artificially high values of FRESCO effective cloud fraction for those pixels and consequently to artificially low values of FRESCO cloud pressure (see equation 1). On the other side, the B-band is less impacted by the bias albedo and so is the FRESCO-B cloud pressure. This albedo bias impacting more the $O_2$ A-band measurements than the $O_2$ B-band leads to a higher cloud pressure difference than expected.

The mean difference of cloud pressure between FRESCO-B and FRESCO is $11.5\,\mathrm{hPa}$, but the difference does not have

the same signification according to the underlying surface: over ocean, the two pressures indicate different pieces of information about the vertical structure of the cloud layer, while over vegetation, the difference is partly due to the difference of surface albedo. Over vegetation, the FRESCO-B cloud pressure is more accurate than the FRESCO one. In the future we also recommend to use a surface albedo database which takes into account the anisotropy of the surface to avoid biases.





## 5.2 Comparison with ground-based measurements

### 5.2.1 Cloudnet target classification product

Cloudnet is a network of ground-based measurement facilities for the evaluation of clouds and aerosols in forecast models. Cloudnet started around 2001 with three observations sites (at Cabauw, Palaiseau and Chilbolton) and includes now five other

permanent sites (at Juelich, Leipzig, Lindenberg, Mace Head and Potenza). In this study, we reject the data from the Mace Head site as this one is on the seaside, while the other sites can be considered as surrounded by vegetation. These sites are equipped with active sensors such as lidar and Doppler millimeter-wave radar that provide vertical profiles of cloud and aerosol properties as well as ice and liquid cloud water content at high temporal and spatial resolution. In this study, we use the Cloudnet Level 2 classification product (Illingworth et al., 2007) which is based on the combination of the vertically pointing Doppler cloud

radar and backscatter lidar, and is available approximately every 30s. This product classifies each vertical layer as 1 of 11 classes, which distinguishes ice and water clouds, precipitation, aerosols, insects, clear sky and combination thereof. Indeed the radar detects large particles such as rain and drizzle drops, ice particles and insects, while the lidar is sensitive to smaller particles such as cloud droplets and aerosols. The target classification product also contains cloud top height and cloud base height. Cloud top and cloud base heights correspond respectively to the highest and lowest altitudes of the backscatter altitude

grid boxes, that have clouds. Consequently, for multilayer cloud situations, the cloud top height refers to the top of the highest layer, while the cloud base height refers to the base of the lowest one.

### 5.2.2 Methodology

In this part, we perform comparisons between the two versions of FRESCO and the Cloudnet target classification product and cloud boundaries for the seven Cloudnet observations sites in July and August 2014. For every GOME-2B pixel collocated

with a Cloudnet site, we select 1 hour (+/-30 min) of Cloudnet target classification. For every cloud height measurement from GOME-2B, there are about 120 (temporal) × 495 (vertical) radar/lidar backscatter Cloudnet pixels, which are classified as one of the 11 categories mentioned earlier. In this study, we keep the cloudy cases, which consist of the classes "ice" and "cloud droplets only" as well as the precipitations ("drizzle or rain", "drizzle or rain and cloud droplets", "ice and supercooled droplets", "melting ice", "melting ice and cloud droplets"). We then determine the height distributions of the backscatter pixels,

from 270 to 15000 m with a bin size of $270\,\mathrm{m}$, following the method defined by Wang and Stammes (2014). If the distribution presents a unique mode, without an interruption by a clear-sky pixel, the cloud is considered as a monolayer case. Otherwise, we classify it as a multilayer case. The Cloudnet cloud top and base heights are calculated averaging the cloud top and base heights in the 1-hour period around GOME-2B overpass time.

### 5.2.3 Results

For the two considered months, the collocation process provides us 339 collocated cloud cases, that we further filter as follows:



- We keep only the cases for which the Cloudnet cloud fraction is higher than 0.05. This Cloudnet cloud fraction is calculated dividing the number of cloudy pixels by the number of pixels accumulated during the one hour period. With this filtering, we exclude the almost cloud-free scenes.

- Then we further filter the Cloudnet data excluding the cases for which the standard deviation of the cloud top height exceeds 1.5 km. This criterion used in Veefkind et al. (2016) allows to avoid the cases with a large temporal variability during the satellite overpass.

- Finally, we filter the cases according to the FRESCO-B cloud fraction, excluding the cases with $c < 0.1$. Indeed, it is known that the FRESCO cloud pressures are often too low when the cloud fraction is lower than 0.1 (Wang et al., 2008).

Those criteteria leave us with 138 cases: monolayer clouds are represented in panel (a) of Figure 9, while multilayer situations are shown in panel (b). On both panels, the clouds are ordered by increasing Cloudnet top height altitude. We can distinguish three different regimes, for which the clouds properties are resumed in Table 7 :

1. For 68 cases (49% of the clouds), the FRESCO-B cloud pressure is inside the Cloudnet cloud boundaries. This population correponds to 46% of the monolayer clouds and to 59% of the multilayers clouds and concern vertically extended middle to high clouds (mean cloud height of 5.5 km and mean cloud thickness of 3.7 km). With FRESCO, the retrieved pressure is inside the Cloudnet boundaries for 64 cases (46% of the clouds) that are a bit higher (mean altitude of 6 km) and a bit thicker (mean thickness 4.1 km) than for FRESCO-B.

2. In 51 cases, the FRESCO-B cloud pressure indicate an altitude lower than the Cloudnet cloud boundaries. This expected behaviour concerns mainly middle to high (mean cloud height of 5.7 km) monolayer clouds with a limited vertical extent (mean geometrical thickness of 2 km). As already shown (Wang et al., 2008; Sneep et al., 2008), light can penetrate within the clouds where it is absorbed. As this phenomenon is not taken into account in FRESCO, the retrieved cloud height is lower than the cloud top height. This is the case in other retrieval algorithms based on oxygen absorption (Ferlay et al., 2010; Desmons et al., 2013). FRESCO cloud altitude is lower than Cloudnet for 53 cases. It concerns lower (mean altitude of 5.1 km) and thinner (mean thickness of 1.6 km) clouds than for FRESCO-B.

3. In the last 19 cases, the FRESCO-B cloud pressure indicate an altitude higher than the Cloudnet cloud boundaries. This concerns mainly monolayer low (mean top height of 1.9 km) thin (mean cloud geometrical thickness of 0.4 km) clouds. In those cases, many photons are transmitted and reach the surface before being reflected back to space. This situation concerns 21 cases for FRESCO, also quite low (mean top height of 2.3 km) and thin (mean cloud geometrical thickness of 0.5 km).

The results shown by FRESCO-B are very similar to the ones obtained with FRESCO, however FRESCO-B performs slightly better than FRESCO as with FRESCO-B, 49% of the retrieved cloud pressures are within the Cloudnet cloud boundaries, while this number is 46% with FRESCO. Also for monolayer clouds, FRESCO-B retrieves an altitude within Cloudnet range in 46% of the cases, while this value is 41% with FRESCO. For multilayers clouds, FRESCO performs slighlty better than




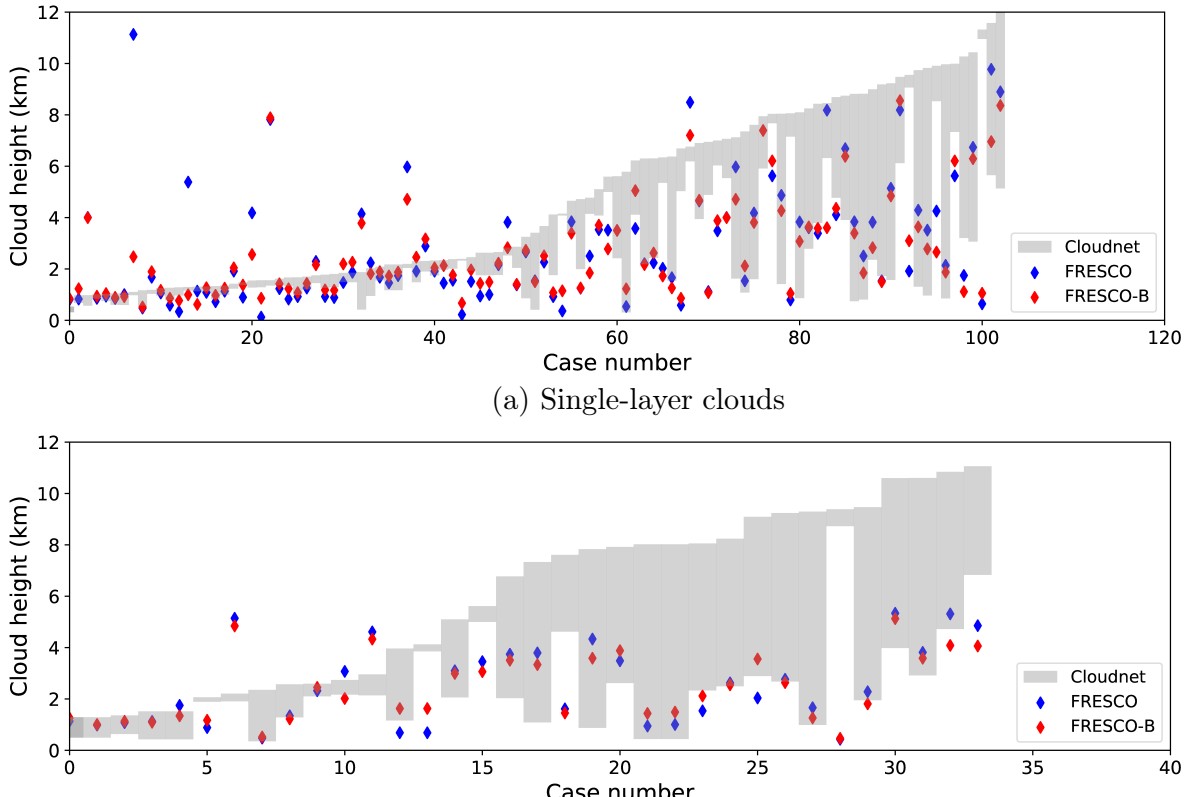

(a) Single-layer clouds

**Figure 9.** Comparison of the cloud heights retrieved by FRESCO-B, FRESCO and Cloudnet in July and August 2014, for six Cloundet sites (Hyytiala, Juelich,Leipzig, Lindenberg, Palaiseau and Potenza).

FRESCO-B, indicating an altitude within Cloudnet range for 62% of the cases, while this value is 59% with FRESCO-B. Those performances are very promising and show that it would be very valuable to use both FRESCO-B and FRESCO retrievals, particularly in the case of multilayers clouds.

Like in the previous section, we have also filtered the pixels according to their position in the swath, keeping only the nadir and eastern pixels of the swath. Then, the FRESCO-B cloud pressure is inside the Cloudnet cloud boundaries in 49% of the cases, as with all the pixels, for FRESCO, this number stays stable too as it is now 47 % (46% for all the swath). Excluding the western pixels of the swath doesn't seem to change the performances of the two algorithms. However, the size of the database is too small to deduce strong conclusions. Also, we should keep in mind that the Cloudnet sites are situated in Europe, while the study made by Lorente et al. (2018) takes place in Amazonia. The vegetation is probably very different between those two regions (grass/forests) and so are their surface albedos.



| | Collocated cases | Inside CN. range | | Lower than CN. altitude range | | Higher than CN. altitude range | |
|---|---|---|---|---|---|---|---|
| FRESCO-B | 138 | 68 (49%) | $\overline{z_{CN}}$=5.5 km $\overline{h_{CN}}$=3.7 km | 51 (35%) | $\overline{z_{CN}}$=5.7 km $\overline{h_{CN}}$=2.0 km | 19 (15%) | $\overline{z_{CN}}$=1.9 km $\overline{h_{CN}}$=0.4 km |
| FRESCO | 138 | 64 (46%) | $\overline{z_{CN}}$=6 km $\overline{h_{CN}}$=4.1 km | 53 (38%) | $\overline{z_{CN}}$=5.1 km $\overline{h_{CN}}$=1.6 km | 21 (16%) | $\overline{z_{CN}}$=2.3 km $\overline{h_{CN}}$=0.5 km |

**Table 7.** Number of collocated cloud cases for which FRESCO-B and FRESCO cloud pressures are inside, upper or higher than the Cloudnet cloud pressure range. For each case, the mean Cloudnet cloud altitude ($\overline{z_{CN}}$) and geometrical thickness ($\overline{h_{CN}}$) are indicated. Collocated Cloudnet/GOME-2B for July and August 2014 for the Cloudnet sites of Hyytiala, Juelich,Leipzig, Lindenberg, Palaiseau and Potenza.

## 6 Conclusions

In this paper, we present a new cloud retrieval algorithm called FRESCO-B, for GOME-2 measurements in the oxygen B-band. FRESCO-B is based on the algorithm FRESCO, which uses measurements in the oxygen A-band to retrieve cloud properties (effective cloud fraction and cloud pressure). However, while the surface albedo in the $O_2$ A-band is very low over ocean,

the albedo takes larger values over vegetation, which can lead to biases in the retrievals. In this paper, we apply FRESCO-B to GOME-2B measurements in the $O_2$ B-band, which is a wavelength range where the surface albedo stays relatively low whatever the underlying surface is.

First, we have simulated cloudy scenes which have shown that FRESCO-B and FRESCO retrievals indicate an altitude inside the cloud layer but well below the top, which was expected. We have noted that the difference between the pressures obtained

with the two algorithms depends on the geometry of the scenes and ranges between -10 hPa et +10 hPa, excepted for high thin clouds where it can reach +30 hPa. Then, inter-comparisons between FRESCO-B and FRESCO over one month of GOME-2B data have shown that the effective cloud fraction retrieved is very similar in the two bands. These comparisons have also revealed that FRESCO-B retrieves a higher cloud pressure than FRESCO (mean difference of 11.5 hPa), and is more accurate over vegetation ($\chi^2$ is lower over vegetation than for other surfaces). Finally, we have validated FRESCO-B and FRESCO to

in-situ data over vegetation obtained with the Cloudnet network of instruments. These comparisons have shown that FRESCO-B and FRESCO can retrieve a pressure which stands inside the cloud layers for cloud that are not too far from the Lambertian model. In the future, we would like to apply FRESCO-B and FRESCO to TROPOMI (Veefkind et al., 2012) measurements in the $O_2$ A and B-bands. Indeed its high spatial resolution (7 km × 3.5 km) compared to previous spectrometers (80 km × 40 km for GOME-2B) should provide much more detailed cloud structures.

In the future, the authors would like to merge the measurements in the $O_2$ A and B-bands in order to retrieve more information about the vertical structure of cloud layers. Excluding the western part of the swath of GOME-2B in the comparisons have improved the performances of FRESCO-B, which corroborates the conclusions of Lorente et al. (2018) on the surface albedo



bias for these pixels. Since this bias leads to biases in cloud retrievals, the authors recommend to take into account the anisotropy of the surface reflectances for future surface albedo climatologies (on-going work within AC SAF).

*Acknowledgements.* This research was funded by the Netherlands Space Office (NSO) User Support Programme Space Research through the FRESCO-B project, ALW-GO/16-23. We acknowledge the EU Cloudnet and the EU H2020 project ACTRIS projects for providing the cloud classification datasets for the Cloudnet sites. We thank Maarten Sneep and Olaf Tuinder for the processing and visualization tools they have provided to the authors.




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
