# Peer review of "FRESCO-B: A fast cloud retrieval algorithm using oxygen B-band measurements from GOME-2"

_Atmospheric Measurement Techniques, 2018_

## Referee Comment (RC1) · Anonymous Referee #1 · 29 Dec 2018

Review comments on manuscript "FRESCO-B: A fast cloud retrieval algorithm using oxygen B-band measurements from GOME-2"

Authors: M. Desmons, P. Wang, P. Stammes, and L. G. Tilstra MS No.: amt-2018-420 MS Type: Research article

Comments:

This paper presents the new FRESCO-B algorithm, which retrieves cloud properties using measurements at the oxygen B-band regions. This new algorithm adopts the same infrastructure of the robust and well-tested FRESCO model. The paper gives the theoretical basis, radiative transfer simulations and performance evaluation. The

work is helpful in further understand the cloud retrievals with oxygen A- and B-bands. The paper is well written and the topic is suitable for publication in AMT. I recommend publication after some minor revisions. Some comments are as follows:

1) Trying to understand the simulation results shown in Figure 6. Using Fig 6a as an example (it's easier to analyze since the surface albedo is the same for both FRESCO-B and FRESCO), why do you think the pressure difference is negative for the thin-and-low cloud case? There are three factors that contribute to the pressure difference: (a) the photon path differences inside clouds; (b) the photon path differences below clouds; and (c) the differences in the effective cloud fraction. The only one that can cause negative pressure difference is (c). Since FRESCO-B has a slightly larger effective cloud fraction, the total photon path for the clear part of the pixel should be smaller than FRESCO. I'm thinking if you force the effective cloud fraction to be the same, the negative cloud pressure difference will probably disappear for Fig6a.

Specific comments:

P5 Fig 3: Please check the vertical axis of the figures. It doesn't look right.

P18 Fig 9: Missing the (b) panel caption

––––––––––––––––––––––––––––––––

---

## Referee Comment (RC2) · Anonymous Referee #2 · 7 Jan 2019

This paper presents a new FRESCO-B algorithm that was built up on the FRESCO algorithm but uses spectral measurements in the O2 B-band, whereas FRESCO algorithm uses the O2 A-band measurements. Both FRESCO-B and FRESCO algorithms retrieve cloud effective fraction and cloud top pressure from three 1-nm wide bands around the corresponding O2 absorption bands. The development of FRESCO-B algorithm is motivated for its valuable application over vegetation surface that has much lower surface albedo in the O2 B-band. This work demonstrated the promising retrievals using the FRESCO-B algorithm with both the synthetic data and the GOME-2 measured data. The study is well designed and matches the scope of AMT. The manuscript is well written. I only have a few minor comments as below.

(1) For the algorithm, why select only three 1-nm wide windows instead of performing spectral fitting at all wavelengths of GOME-2 measurements around O2 A- and B-band? Is it supposed to have more information if more wavelengths are used?

(2) Can authors include the definition for "effective cloud fraction" and explain why cloud effective fraction can exceed 1 (Figure 5)?

(3) Figure 5 and 6. Need to label the left and right panels with (a) and (b), as these are indicated in the figure caption.

(4) Page 7 line 15: "Figure 4b" -> "Figure 5b".

(5) Table 2&3: As shown in the simulated retrievals in section 4, cloud effective fraction is larger from O2-B retrievals. Why the differences are negative for Land and Vegetation cases of GOME-2 retrievals? Ok, I found this is discussed on page 10.

(6) Figure 7a is not discussed in the text.

(7) It seems to me Figure 7b shows substantial land areas with negative cloud pressure difference. However, global average of this difference over Land is positive in Table 4, which may not consistent with Figure 7b. Please double check.

(8) In Figure 7, it appears some correlation may exist between the differences in effective cloud fraction and the differences in cloud top pressure. For instance, areas over land with negative cloud pressure difference tend to have negative cloud fraction difference. It seems a low bias in cloud fraction may lead to low bias in cloud pressure?

(9) Figure 9: Need to add label "(b) Multi-layer clouds"
* * *

---

## Author Comment (AC1) · 4 Mar 2019

***Response to Anonymous Referee #1 comments on*** **"FRESCO-B: A fast cloud retrieval algorithm using oxygen B-band measurements from GOME-2"** ***by*** **Marine Desmons et al.**

Authors: M. Desmons, P. Wang, P. Stammes, and L. G. Tilstra

The authors are grateful to the referee for the constructive evaluation and useful comments on the paper. In the following, a point-by-point reply is given, with the Referee's comments in italics.

*This paper presents the new FRESCO-B algorithm, which retrieves cloud properties using measurements at the oxygen B-band regions. This new algorithm adopts the same infrastructure of the robust and well-tested FRESCO model. The paper gives the theoretical basis, radiative transfer simulations and performance evaluation. The work is helpful in further understand the cloud retrievals with oxygen A- and B-bands. The paper is well written and the topic is suitable for publication in AMT. I recommend publication after some minor revisions. Some comments are as follows:*

*1) Trying to understand the simulation results shown in Figure 6. Using Fig 6a as an example (it's easier to analyze since the surface albedo is the same for both FRESCO- B and FRESCO), why do you think the pressure difference is negative for the thin-and- low cloud case? There are three factors that contribute to the pressure difference: (a) the photon path differences inside clouds; (b) the photon path differences below clouds; and (c) the differences in the effective cloud fraction. The only one that can cause negative pressure difference is (c). Since FRESCO-B has a slightly larger effective cloud fraction, the total photon path for the clear part of the pixel should be smaller than FRESCO. I'm thinking if you force the effective cloud fraction to be the same, the negative cloud pressure difference will probably disappear for Fig6a.*

This is a very interesting observation by the reviewer: for the case of a thin low cloud the FRESCO-B pressure is a bit lower than the FRESCO pressure. This holds for both the ocean and vegetation cases, except for the most oblique geometry for vegetation. In addition to the three factors mentioned by the reviewer there is also the factor of the light path above the cloud, which could be influenced by reflection by the cloud. Since the B-band is weaker than the A-band, multiple scattering between the cloud particles and the molecular Rayleigh scatterers above, inside, and below the cloud is stronger in the B-band than in the A-band. At 685 nm there is 50 % more Rayleigh scattering than at 760 nm. Most Rayleigh scattering is above 5 km, so by scattering above the cloud the B-band pressure would be lower than the A-band pressure.

The effective cloud fraction appears to be slightly larger in the B-band than in the A-band, which is well observed by the reviewer. This effect would counteract the above effect, since a larger cloud fraction means that more photons come from the cloud level instead of the clear sky. We followed the suggestion of the reviewer, and forced the effective cloud fraction in the B-band to be equal to that in the A-band. We obtain the same negative pressure difference. The values are resumed in the following table:

| $c_{eff}$=0.4679 | $P_{FRESCO-B}$=685.285 hPa |
|---|---|
| | $P_{FRESCO}$=689.318 hPa |
| $c_{eff}$=0.4558 | $P_{FRESCO-B}$=685.067 hPa |
| | $P_{FRESCO}$=689.139 hPa |

We changed the manuscript accordingly, section 4.3, p19:

...For the case of a thin low cloud, the FRESCO-B pressure is a bit lower than the FRESCO pressure. This holds for both the ocean and vegetation cases, except for the most oblique geometry for vegetation. This feature can be due to the Rayleigh scattering. Since the B-band is weaker than the A-band, multiple scattering between the cloud particles and the molecular Rayleigh scatterers above, inside, and below the cloud is stronger in the B-band than in the A-band. At 685 nm there is 50 % more Rayleigh scattering than at 760 nm. Most Rayleigh scattering is above 5 km, so the pressure retrieved in the B-band is lowered by the scattering happening above the cloud, leading to a smaller (negative) difference of pressures....

*Specific comments:*
*P5 Fig 3: Please check the vertical axis of the figures. It doesn't look right.*

Thanks for identifying this mistake. In Fig. 3 we plotted the reflectance at top-of-atmosphere, whereas we intended to plot the transmittance at the surface, to clearly show the absorption bands of oxygen. We corrected this. The atmospheric set-up and geometry is now given in the caption.

Here is the new figure:

[Figure]

(a) B-band     (b) A-band

Figure 3: Line-by-line transmittances in the oxygen A (b) and B (a) bands (black lines). The transmittances of the overlapping water vapor lines are represented in red. The calculations are computed using the HITRAN 2016 database (Gordon, 2017) and for a solar zenith angle of 0 °.

*P18 Fig 9: Missing the (b) panel caption*

Added. Thanks for noticing this oversight.

---

## Author Comment (AC2) · 4 Mar 2019

***Response to the anonymous referee #2 comments on*** "FRESCO-B: A fast cloud retrieval algorithm using oxygen B-band measurements from GOME-2" ***by*** **Marine Desmons et al.**

Authors: M. Desmons, P. Wang, P. Stammes, and L. G. Tilstra

The authors are grateful to the referee for the constructive evaluation and useful comments. In the following, a point-by-point reply is given, with the Referee's comments in italics.

*This paper presents a new FRESCO-B algorithm that was built up on the FRESCO algorithm but uses spectral measurements in the O2 B-band, whereas FRESCO algorithm uses the O2 A-band measurements. Both FRESCO-B and FRESCO algorithms retrieve cloud effective fraction and cloud top pressure from three 1-nm wide bands around the corresponding O2 absorption bands. The development of FRESCO-B algorithm is motivated for its valuable application over vegetation surface that has much lower surface albedo in the O2 B-band. This work demonstrated the promising retrievals using the FRESCO-B algorithm with both the synthetic data and the GOME- 2 measured data. The study is well designed and matches the scope of AMT. The manuscript is well written. I only have a few minor comments as below.*

*(1) For the algorithm, why select only three 1-nm wide windows instead of performing spectral fitting at all wavelengths of GOME-2 measurements around O2 A- and B- band? Is it supposed to have more information if more wavelengths are used?*

The use of a selected number of wavelengths in the O2 A-band is one of the key features of the FRESCO method as devised by Koelemeijer et al. (2001). The basic idea is that the A-band (and B-band) contains much dependent information, since it consists of hundreds of lines. The reflectances in the selected three wavelength windows contain nearly all independent information that is available in the $O_2$ A band for instruments with the spectral resolution of GOME-2 (Kollewe et al., 1992, Fischer et al., 1992). Using the entire O2 A-band (or B-band) would unnecessarily slow down the retrieval.

*(2) Can authors include the definition for "effective cloud fraction" and explain why cloud effective fraction can exceed 1 (Figure 5)?*

The definition of effective cloud fraction is now given in Sect. 4.2. p7:

...In Figure 5, we can see that the retrieved effective cloud fraction is sometimes higher than 1. This is due to the principle of the FRESCO algorithm. Indeed, the effective cloud fraction is the part of the pixel that the Lambertian cloud has to occupy to match the observed reflectance while the geometric cloud fraction is the part of the pixel that is covered by the true clouds. The effective cloud fraction is strongly coupled to the choice of the cloud albedo Ac. The choice of Ac=0.8 has been made in the FRESCO algorithm (Koelemeijer et al., 2001, Stammes et al., 2008) for different reasons (correction of trace gases for clouds, consistency of the Lambertian

model, ability to approach the measured reflectivities by simulations), and can lead to effective cloud fractions somewhat higher than 1....

*(3) Figure 5 and 6. Need to label the left and right panels with (a) and (b), as these are indicated in the figure caption.*

Done, we apologize for this omission

*(4) Page 7 line 15: "Figure 4b" -> "Figure 5b".*

Corrected.

*(5) Table 2&3: As shown in the simulated retrievals in section 4, cloud effective fraction is larger from O2-B retrievals. Why the differences are negative for Land and Vegetation cases of GOME-2 retrievals? Ok, I found this is discussed on page 10.*

Ok

*(6) Figure 7a is not discussed in the text.*

This figure is described in paragraph 5.1.1, p10

*(7) It seems to me Figure 7b shows substantial land areas with negative cloud pressure difference. However, global average of this difference over Land is positive in Table 4, which may not consistent with Figure 7b. Please double check.*

We have checked and both the table and the figure are correct. We understand your concern, but there are also many land areas with a positive difference, which balance the areas with a negative difference. We have added the histograms of the cloud effective fraction differences and of the cloud pressures differences over ocean and land in the paper in section 5.1 p13, and changed the text accordingly.

[Figure]

Figure 8: Histograms of the differences of effective cloud fractions (panels a and b) and of cloud pressures (panels c and d) between FRESCO-B and FRESCO for July 2014, over ocean (panels a and c) and over land (panels b and d).}

.... Over land, the cloud pressure difference is also positive on average with a mean value of 6.31±49.1hPa. However, as we can see on Figure 8d, the range of the values is larger than over ocean with a lot of pixels having a negative difference of pressure. This is again due to the high variability of the surface albedos for this type of surface and is coherent with the observations we have made on the simulations. Over land, the coefficient of correlation between the difference of pressures and the difference of cloud effective fractions is 0.523....

*(8) In Figure 7, it appears some correlation may exist between the differences in effective cloud fraction and the differences in cloud top pressure. For instance, areas over land with negative cloud pressure difference tend to have negative cloud fraction difference. It seems a low bias in cloud fraction may lead to low bias in cloud pressure?*

We have checked this point and plot the correlation between the difference of pressures and the difference of effective cloud fraction for different surfaces:

[Figure]

| Over ocean | Over land | Over vegetation |
|---|---|---|
| cc=0.049 | cc=0.523 | cc=0.620 |

Consequently, the correlation between the difference of pressures and the difference of effective cloud fractions is inexistent for ocean pixels and moderate for land and vegetation cases. However, we think that the $O_2$ A- and B-bands react differently to cloud fraction and cloud albedo, which both influence effective cloud fraction, and to cloud vertical structure. These three cloud parameters represent different cloud properties that are not generally correlated. Thus, we think that the hints towards such a correlation are not strong enough.

We have added the value of those correlation coefficients in the paper, section 5.1.2:

.... Over ocean, the coefficient of correlation between the difference of pressures and the difference of cloud effective fractions is 0.0949....(p14)

.... Over land, the coefficient of correlation between the difference of pressures and the difference of cloud effective fractions is 0.523.... (p15)

*(9) Figure 9: Need to add label "(b) Multi-layer clouds"*

Done, thanks for noticing this oversight.